# Analysing the Impact of the Bleaching Process on Wet Spun Hemp Yarn Properties

Simona Tripa [1], Naz Kadınkız [2], Ayesha Kanwal [3], Muhammad Anwaar Nazeer [3,*], Ahsan Nazir [3] , Florin Tripa [1] and Muhammet Uzun [2,*]

1    Department of Textiles, Leather and Industrial Management, Faculty of Energy Engineering and Industrial Management, University of Oradea, 410087 Oradea, Romania; simona.tripa@didactic.uoradea.ro (S.T.); tripa.florinmarin@student.uoradea.ro (F.T.)
2    Department of Textile Engineering, Faculty of Technology, Marmara University, Istanbul 34722, Türkiye; nazkadinkiz@gmail.com
3    School of Engineering and Technology, National Textile University, Faisalabad 37610, Pakistan; ahsan@ntu.edu.pk (A.N.)
*    Correspondence: mnazeer13@ku.edu.tr (M.A.N.); m.uzun@marmara.edu.tr (M.U.)

**Abstract:** Historically, cotton has been regarded as a highly sustainable material; however, thorough research indicates otherwise. The increasing levels of pollution and the need to address climate change have led towards a global search for sustainable alternatives. Plants with comparable chemical compositions, such as hemp, are attracting growing attention. The cultivation of hemp can be done with sustainable methods, thereby making it a viable alternative to cotton. This study investigates the mechanical, physical, and dyeing properties of 100% wet-spun hemp yarn in its natural and bleached state with the objective of incorporating its use in both technical and traditional textiles. Although significant academic literature is available on the properties of cotton, there is a noticeable lack of literature based on wet-spun hemp. This research suggests that the bleaching process positively affects wet-spun hemp yarn, thus making it suitable for use by the textile industries in various applications.

**Keywords:** wet-spun hemp; bleached hemp yarn; sustainable alternatives





## 1. Introduction

As a natural fibre, cotton was once considered the epitome of sustainability due to its biodegradable and compostable properties. Known for its incomparable functional and mechanical and physical properties, cotton is used in various applications and industries [1–3]. Comprehensive research has uncovered several drawbacks to the sustainability of cotton fibre, including its excessive use of land, water, pesticides, and its contamination of water. Although organic cotton doesn't use fertilizers, it cannot be viewed as a universal substitute for conventional cotton because it accounts for less than 1% of all cotton production [4]. The global search for a more sustainable alternative has been initiated due to rising pollution, efforts to combat climate change, and the textile industry's position as the second-most environmentally detrimental sector [5].

Plants with a comparable chemical composition, such as hemp, are being considered promising research subjects due to their lower land, water, and pesticide requirements [6,7]. Hemp fibre falls into the category of bast fibres [8]. It is derived from the hemp plant (Cannabis sativa). It is a member of the Cannabaceae family, the most well-known of which is the Cannabis genus, which consists of two main species: Cannabis sativa and Cannabis indica [9,10]. Each species has unique morphological characteristics, growth patterns, and chemical compositions. These plants can be differentiated by the presence of resinous structures primarily located in their stems, leaves, and flowers. These structures are responsible for the production of active compounds like tetrahydrocannabinol (THC) and cannabidiol (CBD). THC and CBD levels can vary between sativa and indica, with

Cannabis sativa plants having generally higher THC levels and Cannabis indica plants having generally higher CBD levels [11,12]. Due to the presence of such cannabinoids, the cultivation and use of hemp plants can be subjected to legal regulations in some countries, while being completely restricted in others [13–15].

Hemp cultivation can be done using environmentally sustainable practices which may result in positive effects on the soil. In particular, hemp has been observed to enhance the soil's nutrient and water retention capabilities, stimulate microbial activity, and enhance soil structure [16–18]. A water retting or dew retting process can be used to break down the parenchyma cells, after which the stalks are processed in decortication machines [19–21]. The process of retting hemp usually takes 10 to 20 days, which helps to separate fibres from the plant's stalk [22]. Furthermore, these fibres can be produced via two distinct methods: dry spinning and wet spinning [23,24]. For this study, yarn made of wet-spun fibres is used. During the wet spinning process, the fibres undergo a preliminary treatment of immersion in heated water tanks. This step softens the pectin, which facilitates the subsequent drawing and separation of the fibres. Additionally, this treatment helps the development of fibre ribbons which ultimately enhances the strength and durability of the fibres [25–27]. The fibres are then aligned in parallel and twisted into yarn. The yarn can be bleached, which substantially affects variables such as the colour, strength, and uniformity of the yarn. Various chemical processes, such as sodium sulphite, sodium bisulphite, sodium hydroxide with EDTA, boiling with sodium hydroxide after treatment with hydrochloric acid, and boiling in oxalic acid, are used for bleaching. The chemical composition of hemp fibre can be seen in Figure 1, which has been adapted from [28]. Hemp fibres are currently being used in composites, insulation, and building materials. Investigating the mechanical and physical properties as well as the dyeing properties of this fibre and its bleached variant produces valuable findings that can improve the utilisation of hemp in the textile sector for high-value applications.

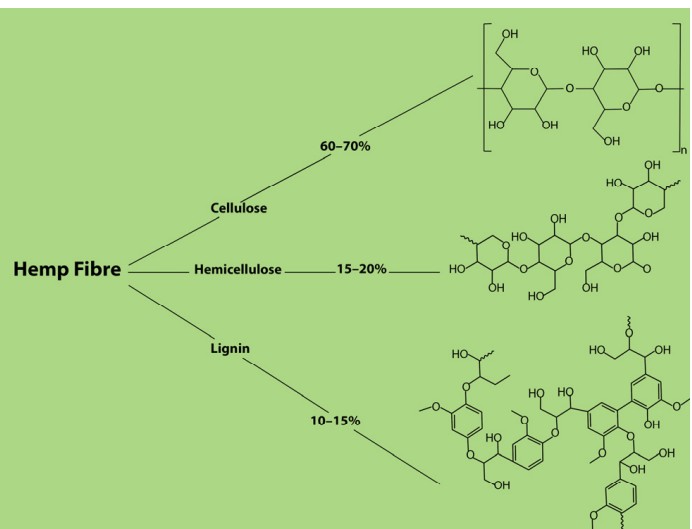

**Figure 1.** The chemical composition of hemp fibre.

The main objective of this study is to investigate the physical, mechanical, and dyeing properties of 100% wet-spun hemp yarn, both in its natural and bleached states, for use in technical and traditional textiles. It is necessary to analyse the mechanical and physical characteristics of the yarns in both their unbleached and bleached conditions to see the modifications in the structure as induced by the process of bleaching. Additionally, within the scope of the study, the natural and bleached variants of the yarns were dyed using reactive dyes, and a comprehensive analysis was conducted to examine the effects of the dyeing process. Hemp fibres are commonly characterized by their natural colouration, which normally ranges from beige to brown to light yellow. The variation in fibre colour is

dependent on the colour of the fibres present in the plant stem, the age of the plant, and the post-harvest processing of the fibres. Based on the preferred hues of the textile industry, five potential yarn colours were selected which were then achieved by the reactive dyeing process and were studied. Although academic work is abundant on the properties of cotton, there is a lack of information when it comes to wet-spun hemp yarn.

## 2. Materials and Methods

### 2.1. Materials

Each test was performed by using 10 replicates of bleached and natural hemp. All tested parameters were conducted under standard test conditions at the temperature of $20 \pm 2$ °C and relative humidity of $65 \pm 4\%$. The yarn replicates were conditioned for 24 h at standard atmospheric pressure with a relative humidity of $65 \pm 2\%$ and a temperature of $21 \pm 1$ °C before the dying process.

Natural and bleached twisted wet-spun hemp yarn was sourced from China. The abbreviations used for the types of yarn are given in Table 1.

**Table 1.** Sample abbreviation.

| Sample Abbreviation | Sample Name |
|---|---|
| N-HEMP | Natural Wet-Spun Hemp |
| B-HEMP | Bleached Wet-Spun Hemp |

Reactive dyes are used for this study due to their uniform colour distribution. The selected dyestuff complies with the OEKO-TEX and GOTS standards [29]. The list of dyestuffs used in this study is given in Table 2. Other ingredients in the dye solution include urea, caustic, and silicates.

**Table 2.** List of dyes.

| Name of Dye | Trade Name of Dye |
|---|---|
| Reactive Yellow | Remazol Yellow SAM |
| Reactive Red | Remazol Red SAM |
| Reactive Blue | Remazol Blue SAM |
| | Remazol Navy RGB 150% |
| Reactive Black | Remazol Map Black NN |

### 2.2. Methods

#### 2.2.1. Yarn Count and Twist Test

The effect of bleach on the yarn count and twist was evaluated on a total of 10 replicates of each yarn. The yarn count test was conducted in accordance with ASTM D1059 standards [30] using a yarn count measuring device, while the twist determination test was carried out using a twist determination device. The experiments were conducted twice and the obtained data was averaged.

#### 2.2.2. Yarn Evenness Test

To investigate the effect of bleach on yarn evenness, a yarn evenness level test was conducted using the Uster Tester 5. The measures include various fundamental characteristics that are crucial for the examination of yarn quality. These criteria include the unevenness percentage (%Um), the coefficient of variation, the mean value coefficient (%CVm), the index, thick places, thin places, neps, and the hairiness of the yarn. Each replicate was tested twice and the data was subjected to statistical analysis. The outcome was presented alongside measures of evenness and standard deviation. The surface appearance of yarn has also been determined using this test.

### 2.2.3. Surface and Morphological Structure

The impact of bleach on the structural properties and quality of yarn was assessed by observing the morphological characteristics of the fibre and yarn using a light microscope. The examination of the 10 replicates involved the analysis of the yarn diameter, the surface properties, and morphological data at various magnification levels. These images were processed using computer-aided image analysis software.

### 2.2.4. Yarn Cross-Section Fibre Count Test

The yarn cross-section fibre count determination test was conducted to evaluate the impact of bleach on the yarn's quality, performance, and durability by measuring the number of fibres it contains. To conduct this test, the yarn is cut across its width and is examined under a microscope to determine the fibre density.

### 2.2.5. Fourier-Transform Infrared Spectroscopy Test

The chemical composition of the yarns was analysed using a Thermo Scientific Nicolet iS50 FTIR spectrometer to detect the effects of bleach. The equipment was sourced from Massachusetts, USA.

### 2.2.6. Thermogravimetric Analysis

The thermal properties of wet-spun hemp yarn variants were examined through the thermogravimetric study (TGA). This study was conducted using the NETZSCH STA 449F3 Jupiter. The equipment was sourced from Selb, Germany. The acquired data was analysed in terms of thermal degradation behaviour and the profile of weight loss.

### 2.2.7. Strength Test

The strength of the yarn was determined through the Instron tensile strength test using the Instron universal testing machine. The 10 replicates were subjected to a constant rate of tension. The test was performed twice on each replicate, and the result was subjected to statistical analysis. The result was presented along with strength value and standard deviation. The tensile strength was again tested using a double-ply yarn to get more precise results with fewer discrepancies caused by variations in fibre thickness. The yarn was twisted at 300 turns per minute (TPM). The yarn twist test was then performed using the Saurer ring twisting machine to examine the increase in strength. The twisting machine was sourced from Arbon, Switzerland.

### 2.2.8. Strength Test of Dyed Yarn

The samples were subjected to reactive dyes for analysis. Table 3 presents the dye formulations used in the reactive dyeing process of the yarns. The dye was combined with hot water and stirred with a magnetic stirrer until it was fully dissolved. Urea was used as an immersion agent. The temperature was carefully controlled during the procedure. Caustic and silicate compounds were added. After achieving homogeneity, the dye solution was introduced into a vat to dye the yarn. Subsequently, the dyed yarn was carefully enclosed in an air-tight nylon bag for a standard waiting time of 20 h. The dyed yarn was technically washed for five minutes to remove any residual dye solution. The strength properties of the dyed yarn were then examined.

### 2.2.9. Colorimetric Data Analysis

ColorTools QC 2.4.3 software was used along with the Datacolor device for the Colorimetric Data Analysis. The L*, a*, and b* values of the dyed yarns were measured three times using a dual-position spectrophotometer under D65 daylight, F11 10, and A 10 light conditions. The primary objective of this test was to present quantifiable data that is not visible to the naked eye. This evaluation offered a foundational understanding of the colour characteristics of wet-spun hemp yarns.

**Table 3.** Dye recipe.

| | % | Trade Name of Dye |
|---|---|---|
| **Grey** | | |
| Reactive Yellow | 0.06 | Remazol Yellow SAM |
| Reactive Red | 0.05 | Remazol Red SAM |
| Reactive Blue | 0.06 | Remazol Blue SAM |
| **Brown** | | |
| Reactive Yellow | 0.5 | Remazol Yellow SAM |
| Reactive Red | 0.26 | Remazol Red SAM |
| Reactive Blue | 0.25 | Remazol Blue SAM |
| **Green** | | |
| Reactive Yellow | 1.3 | Remazol Yellow SAM |
| Reactive Red | 0.5 | Remazol Red SAM |
| Reactive Blue | 1.0 | Remazol Blue SAM |
| **Navy Blue** | | |
| Reactive Yellow | 0.6 | Remazol Ultra Yellow RGB |
| Reactive Red | 1 | Remazol Ultra Red RGB |
| Reactive Blue | 4 | Remazol Navy RGB 150% |
| **Black** | | |
| Reactive Black | 6 | Remazol Map Black NN |

## 3. Results and Discussion

### 3.1. Yarn Count and Twist Test

To evaluate the effect of bleach on the yarn twist, the average twist values of the samples were measured. The findings in Table 4 suggest that bleaching had no impact on the yarn count but it did lead to an increase in the yarn twist. The twist value of hemp had an increase of 5–8%. This result can be attributed to the use of peroxide in the carding process which tightens the voids in the morphology of the fibres, resulting in greater twist values.

**Table 4.** Yarn count and twist values of N-HEMP and B-HEMP.

| Sample Code | Yarn Count (Ne) | T/m |
|---|---|---|
| N-HEMP | 16 | 337 |
| B-HEMP | 16 | 366 |

### 3.2. Yarn Evenness Test

The impact of bleach on the uniformity of the yarn is observed through the assessment of the yarn's evenness. The Um% specifies variation in the yarn thickness or diameter, with a lower value indicating more evenness. The CV% represents the variation in the yarn thickness, with lower values representing less variation. The index represents the overall evenness of the yarn, with a low index signifying improved evenness. The results in Table 5 suggest that the use of bleach has resulted in a partially positive outcome for the hemp yarn. It can be observed by the improved uniformity and a considerable reduction in the irregularities, such as in the thick and thin places of the yarn. However, the neps and hairiness have increased. This may be attributed to the partial modifications to the cellulose structure caused by the bleaching process, which led to alterations in the uniformity of the yarn structure.

**Table 5.** Yarn evenness values of N-HEMP and B-HEMP.

| Sample | Um (%) | CVm (%) | Index (−) | Thin Places (−50%) | Thick Places (+50%) | Neps (+200%) | Hairiness (−) |
|--------|--------|---------|-----------|--------------------|--------------------|--------------|----------------|
| N-HEMP | 26.43 | 33.59 | 4.95 | 4814 | 2949 | 5596 | 1.87 |
| B-HEMP | 19.87 | 25.75 | 3.79 | 1104 | 2204 | 5614 | 2.62 |

*3.3. Surface Appearance of the Yarn*

The surface appearance of yarn is characterised by the fineness value of the yarn. The fineness values of the yarn indicate whether the fibres in the yarn are thin or thick. An abundance of thin places suggests a lack of homogeneity and weakness. Similarly, an abundance of thick places indicates irregularity in the yarn. In Table 6, bleached hemp shows considerably lowered minimum and maximum thickness values, which indicates that the yarn has fewer thin and thick places, specifying a more homogenous structure. The variation in the thickness of N-Hemp can be seen in Figure 2, while the variation in the thickness of B-HEMP can be seen in Figure 3. The sample can be seen in images A, B, and C in Figures 2 and 3 at different magnifications and places.

**Table 6.** Yarn surface thickness of N-HEMP and B-HEMP.

| Thickness Value | N-Hemp | B-Hemp |
|-----------------|--------|--------|
| Minimum (μm) | 204.16 | 123.61 |
| Maximum (μm) | 373.19 | 267.61 |

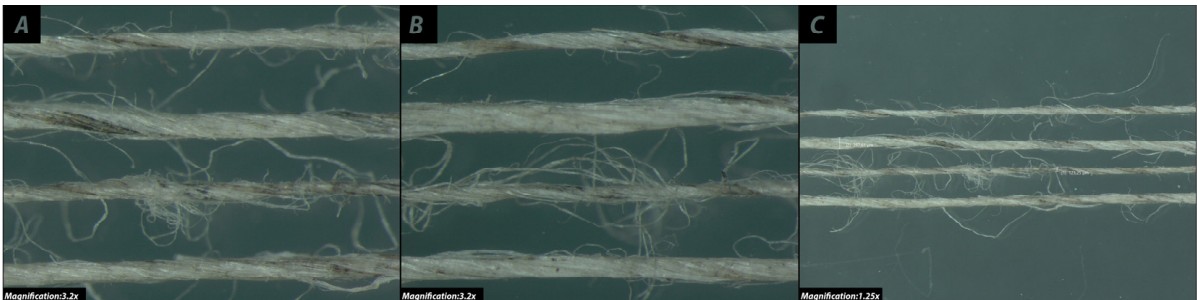

**Figure 2.** The thickness of N-HEMP under the microscope at different places and magnifications of 3.2× (**A**), 3.2× (**B**) and 1.25× (**C**).

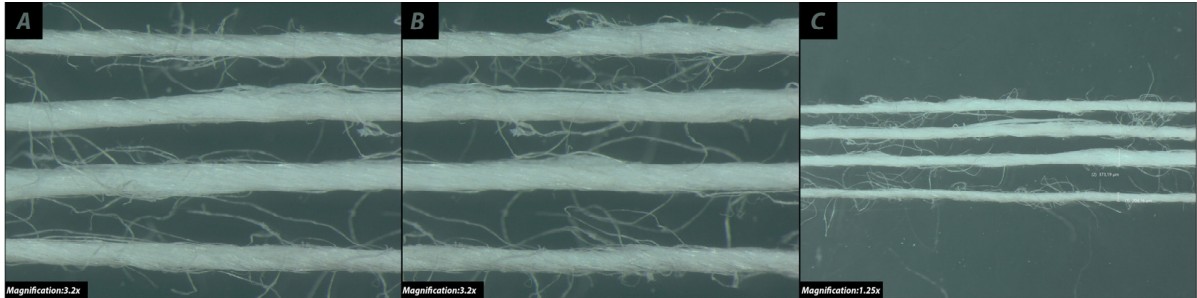

**Figure 3.** The thickness of B-HEMP under the microscope at different places and magnifications of 3.2× (**A**), 3.2× (**B**) and 1.25× (**C**).

*3.4. Surface and Morphological Structure of the Yarn*

By using a light microscope, the effect of bleach on the structural attributes and quality of the yarn was assessed. In the course of fibre development, there are specific stages at which growth ceases. At these specific points, referred to as nodes, fibres gather

crucial nutrients and amorphous regions are formed. Figure 4 confirms when examining the samples under a microscope and using a compensator to bend light, it was noted that the nodal regions displayed colours that fall within the yellow and red spectrum at wavelengths ranging from 700 to 560 nm. In areas with an abundance of crystalline regions and continuous growth, the wavelengths of light ranged from 560 to 400 nm and visual observations indicated the presence of green, blue, and purple hues. Upon examination of the fibre diameters in the horizontal plane, it was obvious that the nodal regions exhibited greater thickness compared to the regions of uninterrupted growth. The strength loss is lower where the amorphous region is more dominant than the crystalline region. Therefore, it can be seen that fractures usually occur in the crystalline regions during strength tests. Moreover, in Figure 5 the microscopic examination of bleached yarns provided visual evidence of the impact of the bleach, signifying the presence of surface irregularities and chemical accumulation on the fibres.

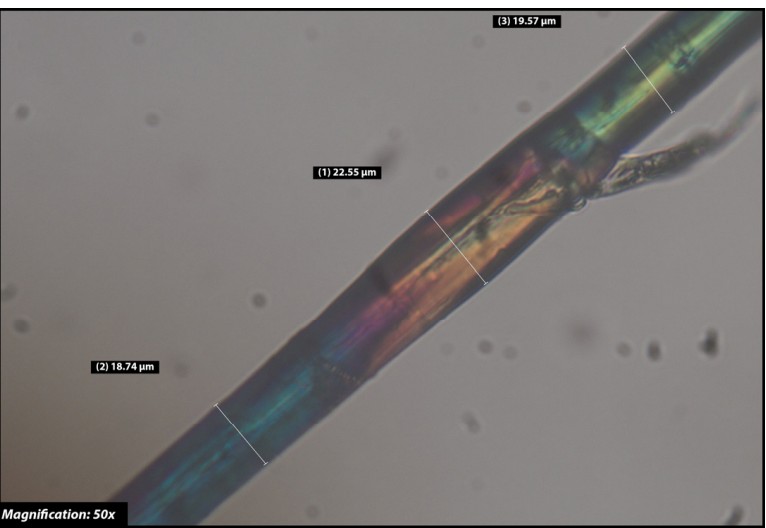

**Figure 4.** Amorphous and crystalline regions of N-HEMP fibre under the microscope.

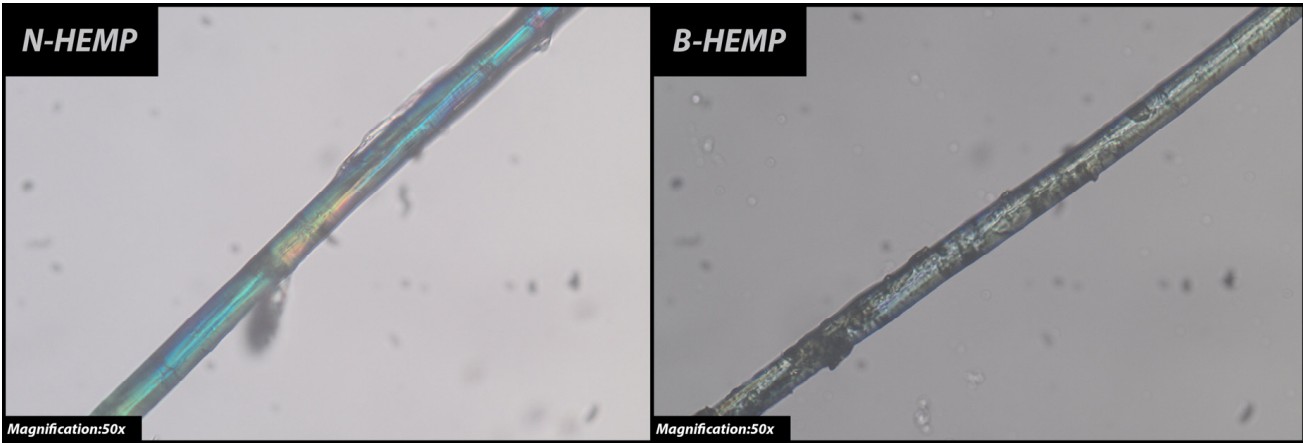

**Figure 5.** Comparison of N-HEMP and B-HEMP under a microscope.

### 3.5. Determination of Fiber Count in Yarn Cross-Section

The findings in Table 7 from the yarn cross-section fibre count determination test indicate that the measured total cross-sectional area of N-HEMP is 35,114.95 $\mu m^2$. Furthermore, the measurements for the diameters of the fibres are 18.24 $\mu m$, 16.23 $\mu m$, 24.43 $\mu m$, and 12.68 $\mu m$, respectively. The presence of different-sized fibres within the yarn has been observed. The average number of fibres per cross-sectional area is 132.50. Figure 6 contains the cross-sectional view of N-HEMP with fibre diameters. According to Table 8,

the measured total cross-sectional area of B-HEMP is 56,944.08 µm² and it is noted that the fibres have varying diameters. The average number of fibres in the cross-section is 190.88. Based on the observed average fibre count of 190.88, it can be concluded that B-HEMP displays a higher fibre density. Figure 7 contains the cross-sectional view of B-HEMP with fibre diameters. The increased abundance of fibres inside the cross-sectional area may indicate a more compact yarn structure, potentially resulting in improved strength properties. However, the notable variations in diameters and the non-uniform dispersion of fibres suggest that the yarn is not homogeneous.

**Table 7.** N-HEMP cross-sectional fibre count.

| Yarn Total Area | Fibre Diameter/Radius (µm) | Fibre Area (µm²) | Average Fibre Count |
|---|---|---|---|
| 35,114.95 µm² | 18.24–9.12 | 261.167 | 132.50 |
| | 16.23–8.115 | 206.779 | |
| | 24.43–12.215 | 468.507 | |
| | 12.68–6.34 | 126.214 | |
| **Average** | 17.895 | 265.5 | |

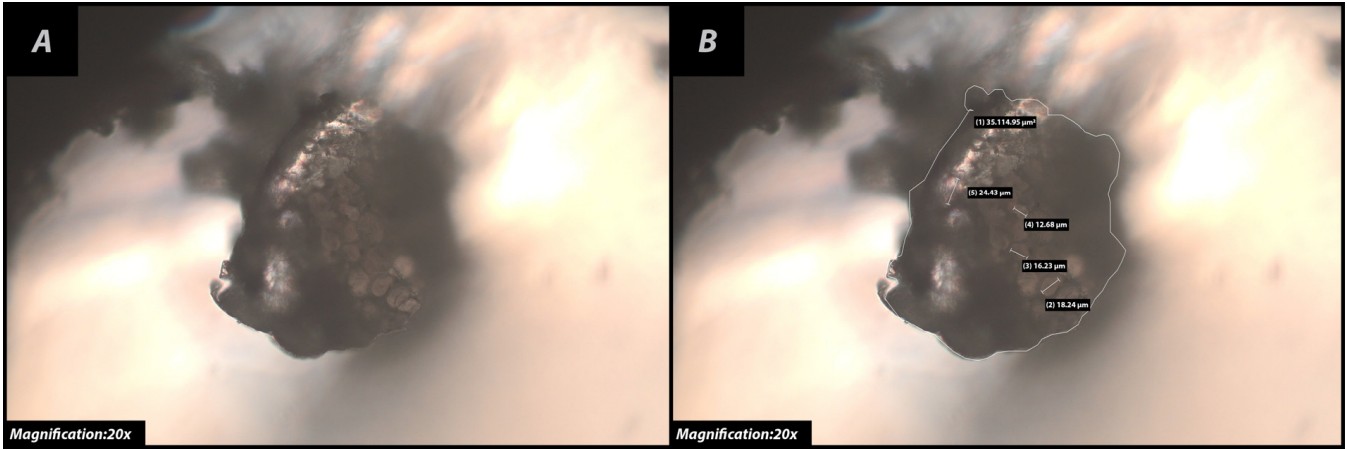

**Figure 6.** (**A**) N-HEMP cross-sectional view without the fibre diameters. (**B**) N-HEMP cross-sectional view with the fibre diameters.

**Table 8.** B-HEMP cross-sectional fibre count.

| Yarn Total Area | Fibre Diameter/Radius (µm) | Fibre Area (µm²) | Average Fibre Count |
|---|---|---|---|
| 56,944.08 µm² | 20.6–10.3 | 333.122 | 190.881 |
| | 25.51–12.755 | 510.84 | |
| | 5.20–2.6 | | |
| | 8.06–4.03 | 50.996 | |
| **Average** | | 298.321 | |

### 3.6. Fourier-Transform Infrared Spectroscopy

FTIR analysis is used to determine the chemical bonds and components of the material. In this analysis, infrared light was used to examine how the material reacts in the infrared region (usually 4000 to 400 cm$^{-1}$). The results in Figure 8 indicate that there was no difference in wavelength range and intensity between N-HEMP and B-HEMP. This specifies that these yarns have the same chemical composition structurally.

### 3.7. Thermogravimetric Analysis

The thermogravimetric study revealed that the thermal decomposition of the N-HEMP initiates at 369.58 °C. At this point, the yarn begins a process of heat deterioration, resulting in a reduction of weight. The temperature at which the yarn shows the greatest weight loss is 513.25 °C. During this stage, the heat deterioration of the yarn becomes more rapid, resulting in additional reduced weight. The residual weight is determined to be 0.624%, representing the proportion of the yarn that remains after thermal degradation. On the other hand, the thermal decomposition of the B-HEMP initiates at 373.25 °C, which is slightly higher than N-HEMP. The temperature at which the highest weight loss occurred is 808.25 °C, which is also higher than the temperature observed for N-HEMP. The residual weight is 0.328%, which is lower compared to the residual weight observed in N-HEMP. Furthermore, this comparison of N-HEMP and B-HEMP is illustrated in Figure 9. The observed rise in the temperature at which initial weight loss occurs suggests that the bleaching process had a positive impact on the thermal stability of the yarn. This observation signifies that the bleached yarn experiences thermal deterioration at elevated temperatures, leading to a comparably substantial reduction in weight. The residual weight of B-HEMP is 0.328%, which suggests that the bleaching process altered the structural composition of the yarn, which led to a decreased number of organic components in the yarn.

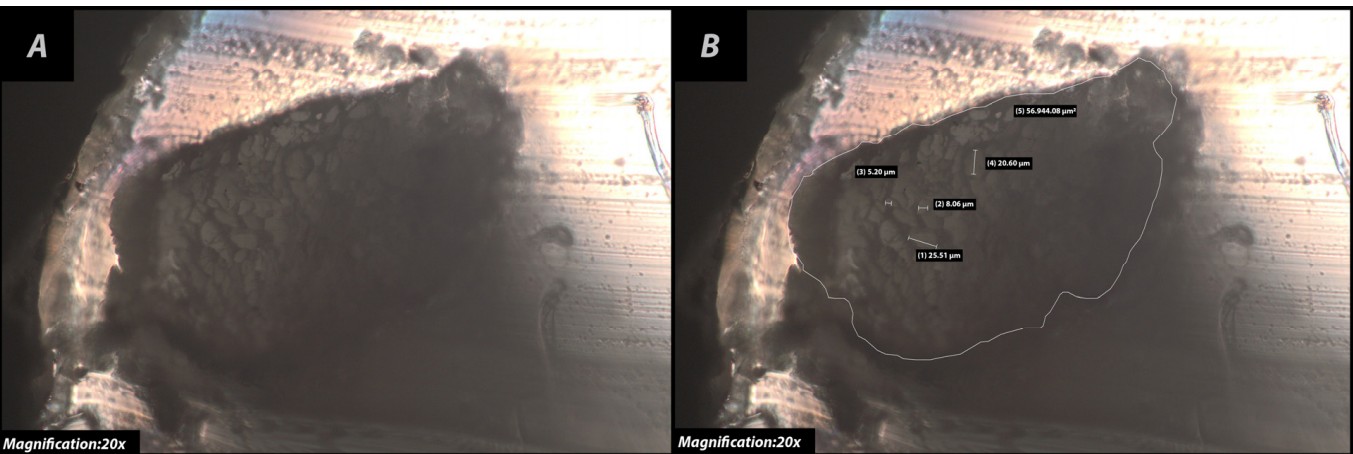

**Figure 7.** (**A**) B-HEMP cross-sectional view without the fibre diameters. (**B**) B-HEMP cross-sectional view with the fibre diameters.

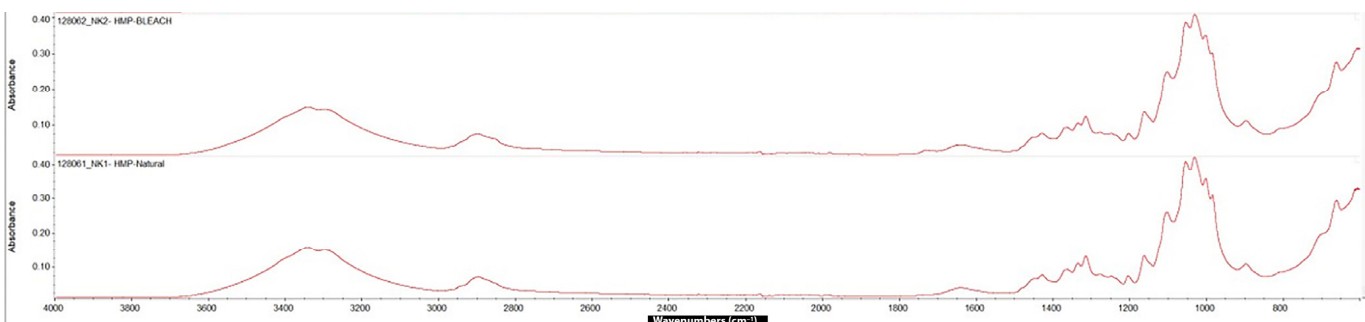

**Figure 8.** FTIR result of N-HEMP and B-HEMP.

### 3.8. Strength Test and the Examination of Breakage Points

The strength test results displayed in Table 9 indicate that the breaking strength, elongation, and tenacity values of hemp yarns display an increase following the bleaching process. This finding suggests that the bleaching process improved the mechanical characteristics, elasticity, and energy absorption of the yarn.

Different breakage points were detected throughout the strength tests conducted on both natural and bleached yarns as shown in Figure 10. During the examination under a microscope, a notched structure was seen at the breakage points. The presence of a notched structure in the cross-section of the yarns during the strength tests suggests that the breakage of the yarns occurred gradually. It was also noted that there were two distinct points of weakness at the breakage points. At one point, a complete breakage occurred instantly, while at the other point, there was a gradual weakening resulting in strength loss. This problem was attributed to inconsistencies in the yarn, resulting from variations in its thickness along the length of the yarn. Microscopic visual inspections were performed to determine the distribution of yarn thickness from a horizontal perspective, allowing the identification of thick and thin places within the yarn. The examination revealed varied thickness distributions and variations in fibre lengths.

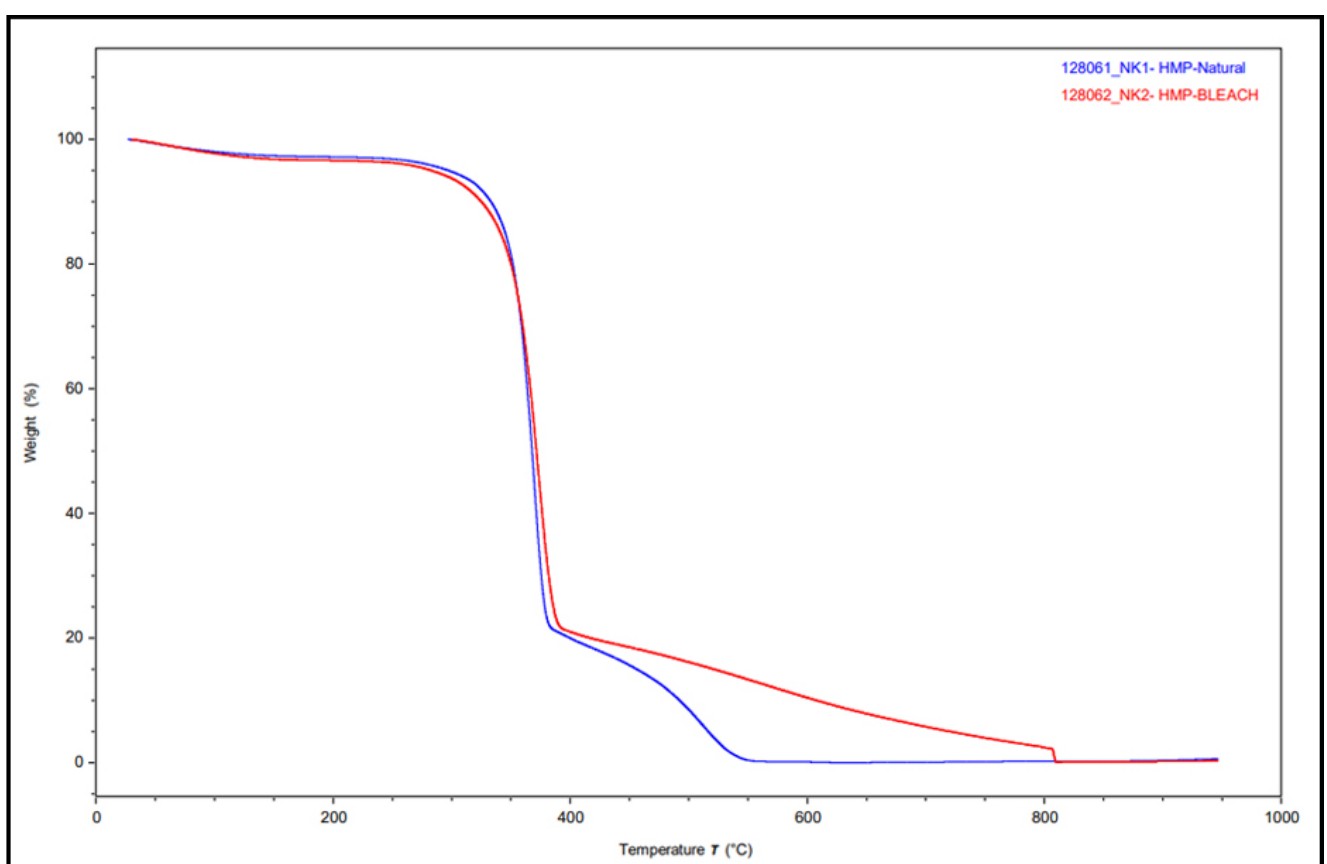

**Figure 9.** Comparison of TGA results of N-HEMP and B-HEMP.

**Table 9.** Strength values of samples.

| Sample | Breaking Strength (N) | Elongation (%) | Tenacity (cN/tex) |
|---|---|---|---|
| N-HEMP | 9.50 | 2.18 | 25.74 |
| B-HEMP | 12.62 | 2.43 | 34.19 |

To obtain more accurate results with fewer inconsistencies caused by variations in fibre thickness, the tensile strength test was repeated using a double-ply yarn. The process involved the double-plying of both 16-count natural and bleached hemp yarns using an Alma Saurer brand ring twisting machine, operating at a speed of 300 TPM. The twisting procedure led to a reduction in inconsistencies. The results in Table 10 indicate that double-plied twisted yarns display a more regular surface appearance with a reduction of the

variation of thin and thick places. The thickness of the double-plied yarn samples can be seen in Figure 11 at different magnifications.

The bleached double-plied wet-spun hemp yarn exhibits superior maximum load and elongation at break values in comparison to the natural double-plied hemp yarn. However, the tenacity values show similarity. Based on the findings presented in Tables 11 and 12, it can be concluded that the bleached double-plied wet-spun hemp yarn shows a notable strength gain.

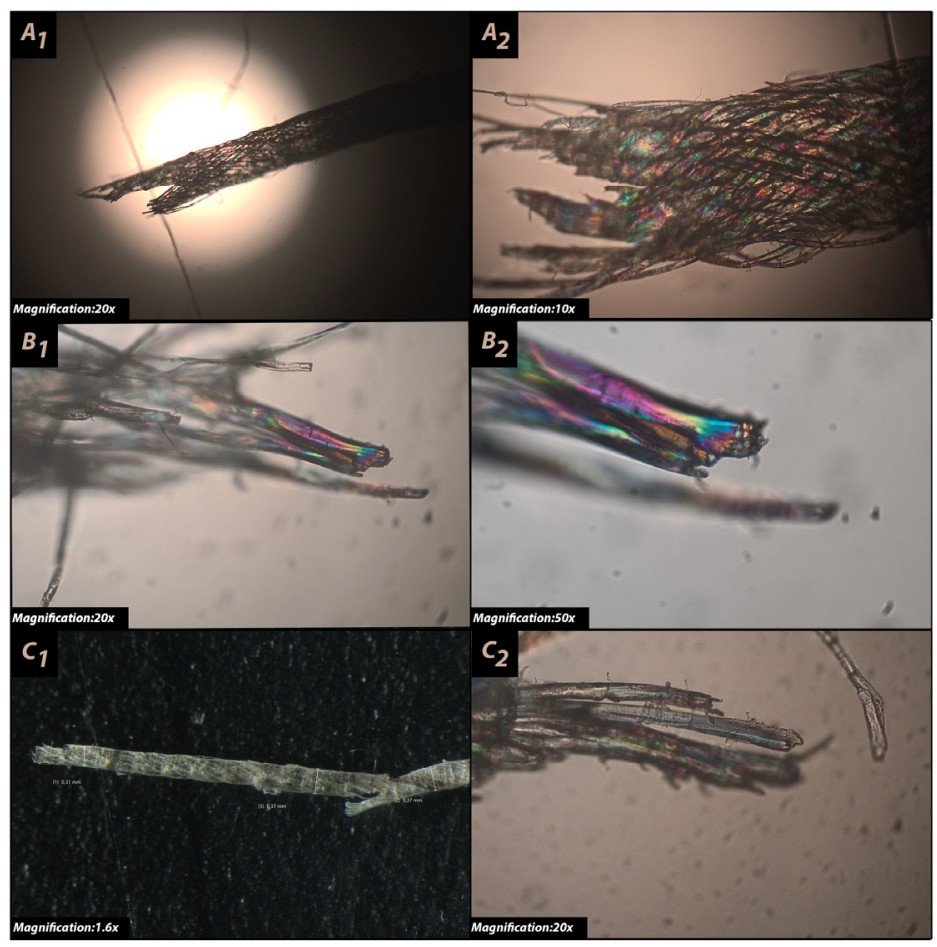

**Figure 10.** ($A_1$,$A_2$) Breakage points of N-HEMP. ($B_1$,$B_2$) Breakage points of B-HEMP fibre. ($C_1$) Breaking of B-HEMP during the strength test. ($C_2$) Breaking of B-HEMP fibre during the strength test.

**Table 10.** The surface thickness of the double-plied yarn.

| Thickness Value | Natural Hemp Double—Ply Yarn | Bleached Hemp Double—Ply Yarn |
| --- | --- | --- |
| Minimum (µm) | 302.80 | 316.00 |
| Maximum (µm) | 352.00 | 415.60 |

According to Table 13, N-HEMP and B-HEMP with a yarn count of Ne 16 show different breaking strength and elongation properties. The average breaking strength value for N-HEMP is 1.41 kgf, while it is 1.28 kgf for B-HEMP. The standard deviation values are 0.15 and 0.09 respectively. On the other hand, the average elongation value of N-HEMP is 2.33%, whereas it is 2.45% for B-HEMP. In the case of 2-Ply yarns with a yarn count of Ne 16×2, the average breaking strength of N-HEMP is 1.21 kgf while it is 1.33 kgf for B-HEMP. The average elongation value of N-HEMP is 2.54% while is 3.02% for B-HEMP.

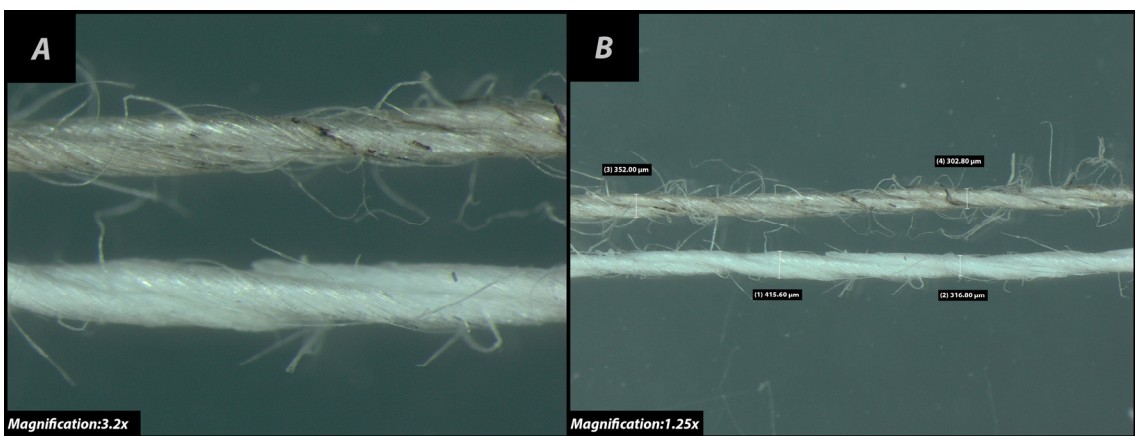

**Figure 11.** Thickness of double-plied samples under a microscope at different magnifications of 3.2× (**A**) and 1.25× (**B**).

**Table 11.** Strength values of natural double-plied hemp yarn.

|  | Maximum Load (N) | Breaking Elongation (%) | Tenacity (%) |
|---|---|---|---|
| Average | 11.87 | 2.54 | 1.56 |
| Standard Deviation | 2.85 | 0.44 | 0.37 |
| Minimum | 7.79 | 1.84 | 1.03 |
| Maximum | 16.23 | 3.23 | 2.14 |

**Table 12.** Strength values of bleached double-plied hemp yarn.

|  | Maximum Load (N) | Breaking Elongation (%) | Tenacity (%) |
|---|---|---|---|
| Average | 13.03 | 3.02 | 1.71 |
| Standard Deviation | 2.42 | 0.48 | 0.32 |
| Minimum | 7.87 | 2.24 | 1.04 |
| Maximum | 16.25 | 4.30 | 2.14 |

**Table 13.** Strength values of twisted and non-twisted samples.

|  | Yarn Count | Breaking Strength (kgf) | | Breaking Elongation (%) | |
|---|---|---|---|---|---|
|  |  | Average | Standard Deviation | Average | Standard Deviation |
| N-HEMP | Ne 16 | 1.41 | 0.15 | 2.33 | 0.20 |
| B-HEMP | Ne 16 | 1.28 | 0.09 | 2.45 | 0.09 |
| N-HEMP 2-Ply Twisted | Ne 16×2 | 1.21 |  | 2.54 |  |
| B-HEMP 2-Ply Twisted | Ne 16×2 | 1.33 |  | 3.02 |  |

### 3.9. Strength Test of Dyed Yarn

The strength test of dyed N-HEMP revealed in Table 14, noticeable variations in strength characteristics across yarn of different colours, including maximum load, elongation at break, and stress under maximum load. Furthermore, it is worth noting that the coefficient of variation values fluctuated among yarns of different colours, thereby suggesting that certain colours possess more consistent strength properties. This may be

due to multiple reasons such as the amount of the dye substance, the level of penetration of the dye, the size and structure of the molecule of the dye and the chemical interaction between the dye and the yarn. The grey-coloured N-HEMP showed the highest maximum load value among the other yarns, with a measurement of 1211.01 cN. Similarly, the grey colour exhibited the lowest coefficient of variation, suggesting that the strength qualities of grey yarns are of higher consistency. The stress and Strain graph of N-HEMP and B-HEMP dyed in grey colour is visualised in Figures 12 and 13.

The strength test results of dyed B-HEMP are presented in Table 15. The yarns with a brown colour showed the highest maximum load value of 1684.89 cN. The highest coefficient of variation among yarns of the same colour suggests a higher level of variability in their strength properties. Further observation revealed that dyed B-HEMP showed superior qualities in comparison to N-HEMP, specifically in the case of grey-coloured yarn. On the other hand, yarns of alternative colours exhibited poorer values in comparison to N-HEMP.

**Table 14.** Strength test values of dyed N-HEMP.

|  | Grey | Brown | Green | Navy Blue | Black |
|---|---|---|---|---|---|
| **Maximum Load (cN)** | | | | | |
| **Coefficient of Variation** | 44.90 | 6.70 | 38.89 | 13.78 | 25.32 |
| **Maximum** | 1211.01 | 855.61 | 1105.71 | 934.58 | 947.75 |
| **Average** | 808.22 | 794.18 | 770.05 | 816.12 | 740.43 |
| **Standard Deviation** | 362.95 | 53.20 | 299.49 | 112.46 | 187.51 |
| **Elongation at Break (mm)** | | | | | |
| **Coefficient of Variation** | 28.74 | 7.01 | 29.50 | 24.64 | 14.46 |
| **Maximum** | 20.67 | 13.48 | 19.14 | 16.26 | 13.79 |
| **Average** | 14.25 | 12.51 | 13.36 | 12.70 | 11.43 |
| **Standard Deviation** | 4.09 | 0.88 | 9.94 | 3.13 | 1.65 |
| **Stress Under Maximum Load (%)** | | | | | |
| **Coefficient of Variation** | 28.28 | 11.55 | 26.80 | 6.40 | 17.82 |
| **Maximum** | 3.79 | 3.29 | 3.3 | 2.81 | 3.16 |
| **Average** | 3.91 | 3.02 | 2.73 | 2.64 | 2.60 |
| **Standard Deviation** | 0.82 | 0.35 | 0.73 | 0.17 | 0.46 |

*3.10. Colorimetric Data Analysis*

Colorimetric Data Analysis was conducted and the colour coordinates deviations through plane analysis were presented in Figures 14–18. While these deviations vary according to colour tones, no significant changes were observed in the samples.

The colour analysis was conducted under different light sources. The DL* values represent the difference in colour brightness, while the Da* and Db* values indicate colour differences along the green/red and blue/yellow axes, respectively. The DC* value represents the difference in chromaticity, and the Dh* value denotes the difference in colour tone. CMC dE value is used as a metric to calculate the overall colour difference. According to the results, there are some differences in the colour values of N-HEMP and B-HEMP under different light conditions.

For grey and brown samples, negligible differences in lightness (DL) were observed, while a significant difference in colour values Da, Db, Dh and Dc and the overall colour difference DE CMC were observed.

For green and navy-blue samples, a different trend was observed as compared to that observed for grey and brown samples. Here, the difference in colour coordinates for both samples was much lower. This was also evident in the overall colour difference having

a maximum value of 0.95, which falls under the acceptable range in industrial colour evaluation practices.

The colour differences as described above were also observed during visual evaluation as depicted in Figure 19. These colour differences could be attributed to the difference in absorption and fixation tendency of the natural and bleached samples. However, the lower difference observed in Navy Blue and Green colours could be attributed to the high dye concentration used for these colours which could have accommodated the difference in dye absorption.

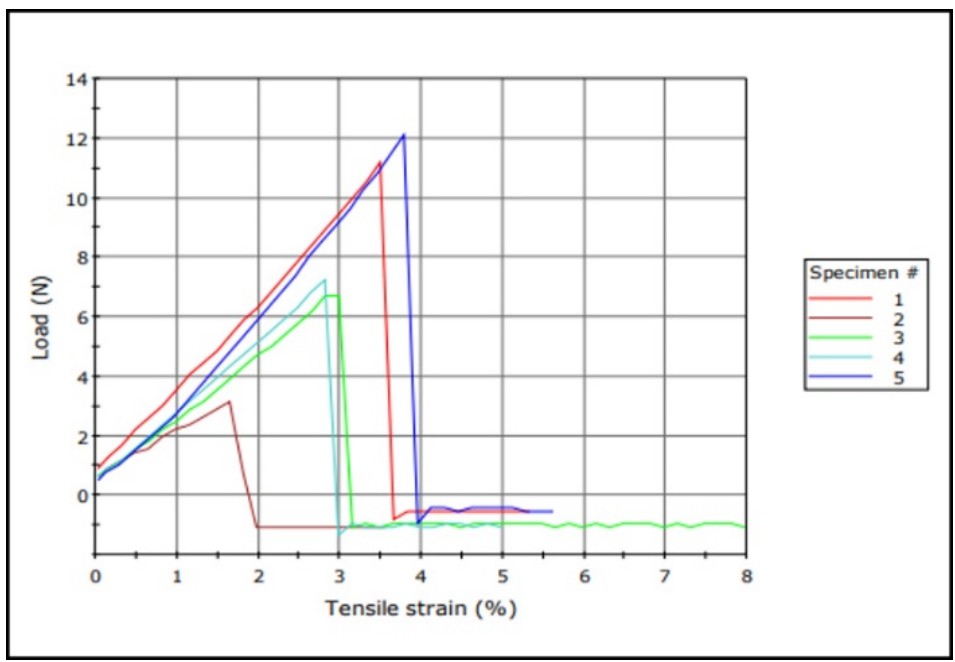

**Figure 12.** Stress-strain graph of N-HEMP dyed in Grey colour.

**Table 15.** Strength test values of dyed B-HEMP.

|  | Grey | Brown | Green | Navy Blue | Black |
|---|---|---|---|---|---|
| **Maximum Load (cN)** | | | | | |
| **Coefficient of Variation** | 10.44 | 15.64 | 18.24 | 11.23 | 14.26 |
| **Maximum** | 1461.11 | 1684.89 | 1461.11 | 1474.28 | 1368.97 |
| **Average** | 1360.20 | 1443.57 | 1270.25 | 1307.55 | 1175.91 |
| **Standard Deviation** | 141.97 | 225.83 | 231.64 | 146.78 | 167.71 |
| **Elongation at Break (mm)** | | | | | |
| **Coefficient of Variation** | 6.91 | 12.98 | 16.37 | 8.97 | 25.21 |
| **Maximum** | 14.90 | 17.31 | 17.72 | 13.72 | 20.22 |
| **Average** | 13.81 | 15.35 | 15.15 | 13.01 | 15.83 |
| **Standard Deviation** | 0.96 | 1.99 | 2.48 | 1.7 | 3.99 |
| **Stress Under Maximum Load (%)** | | | | | |
| **Coefficient of Variation** | 4.27 | 4.53 | 12.42 | 6.87 | 7.80 |
| **Maximum** | 4.15 | 4.33 | 3.96 | 3.82 | 3.66 |
| **Average** | 3.97 | 4.21 | 3.56 | 3.60 | 3.38 |
| **Standard Deviation** | 0.17 | 0.19 | 0.44 | 0.25 | 0.26 |

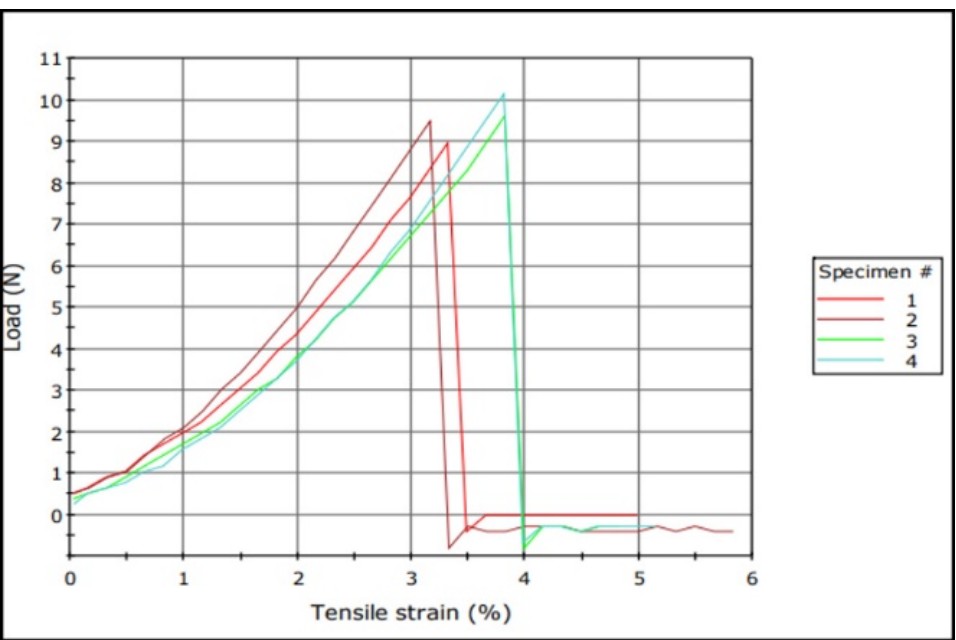

**Figure 13.** Stress/strain graph of B-HEMP dyed in Grey Colour.

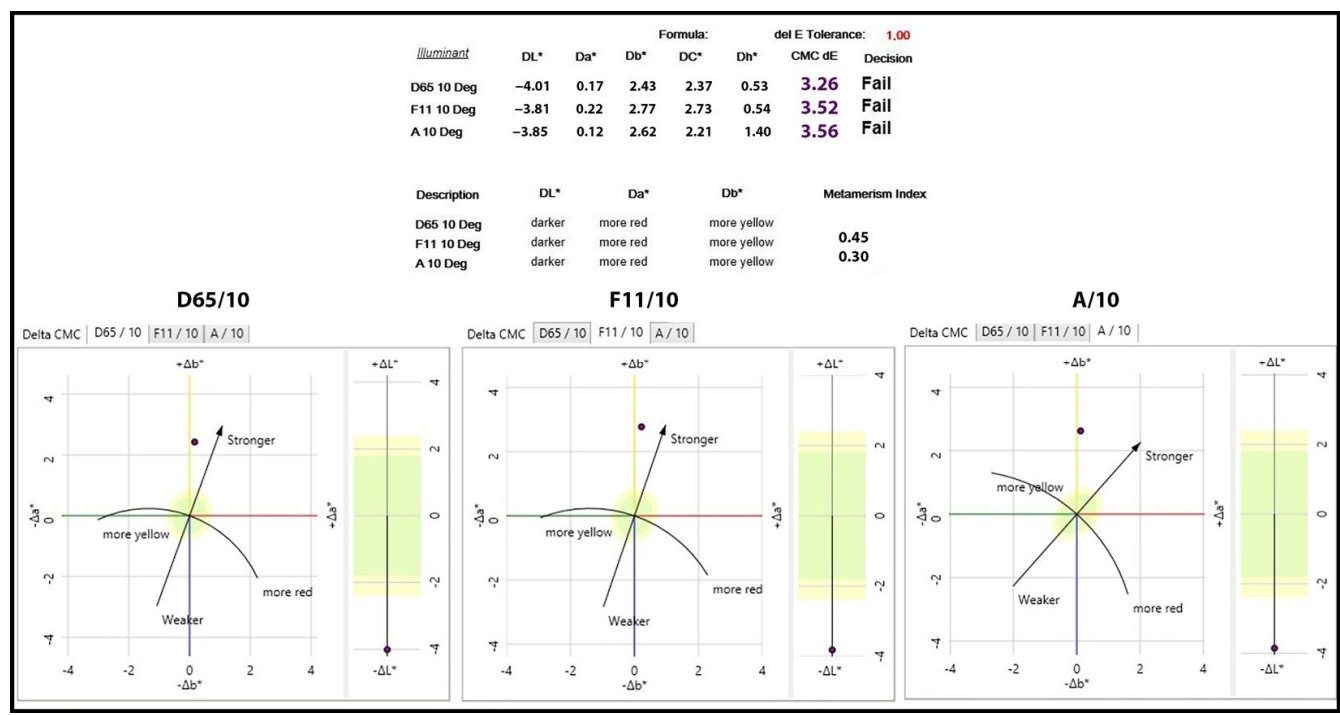

**Figure 14.** N-HEMP and B-HEMP in grey colour.

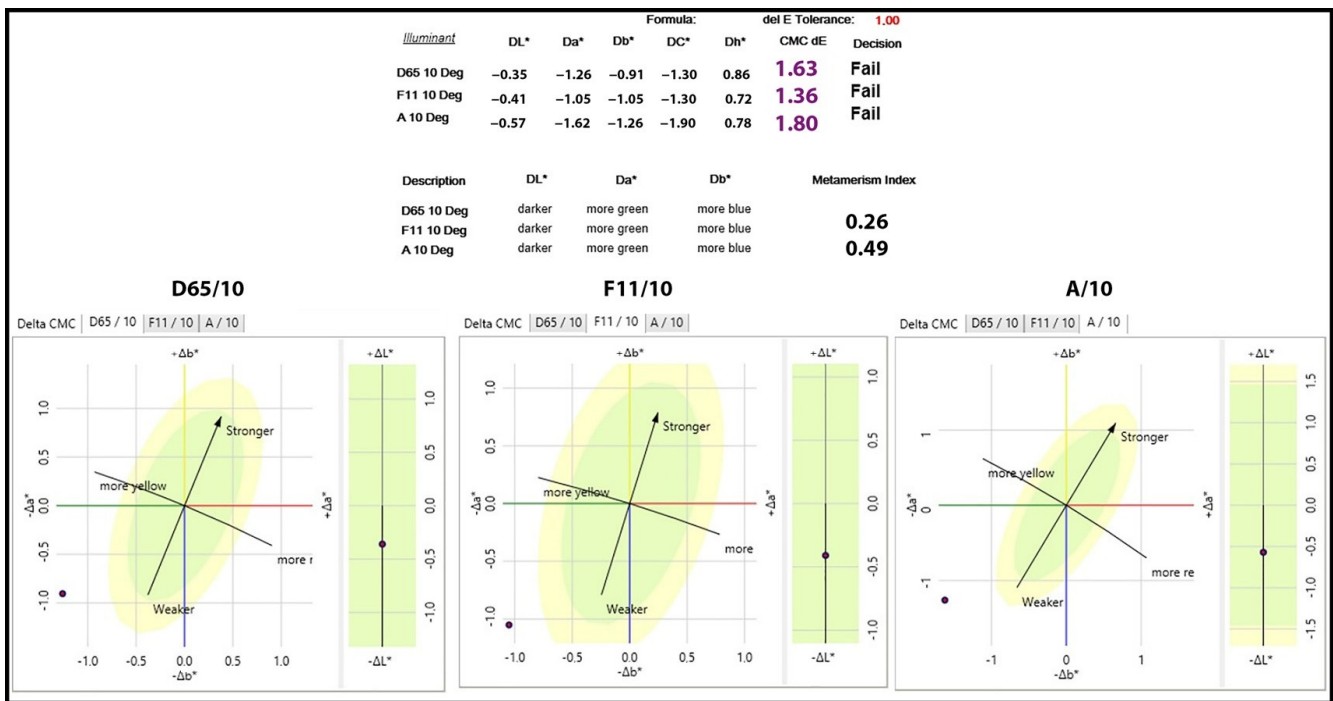

**Figure 15.** N-HEMP and B-HEMP in brown colour.

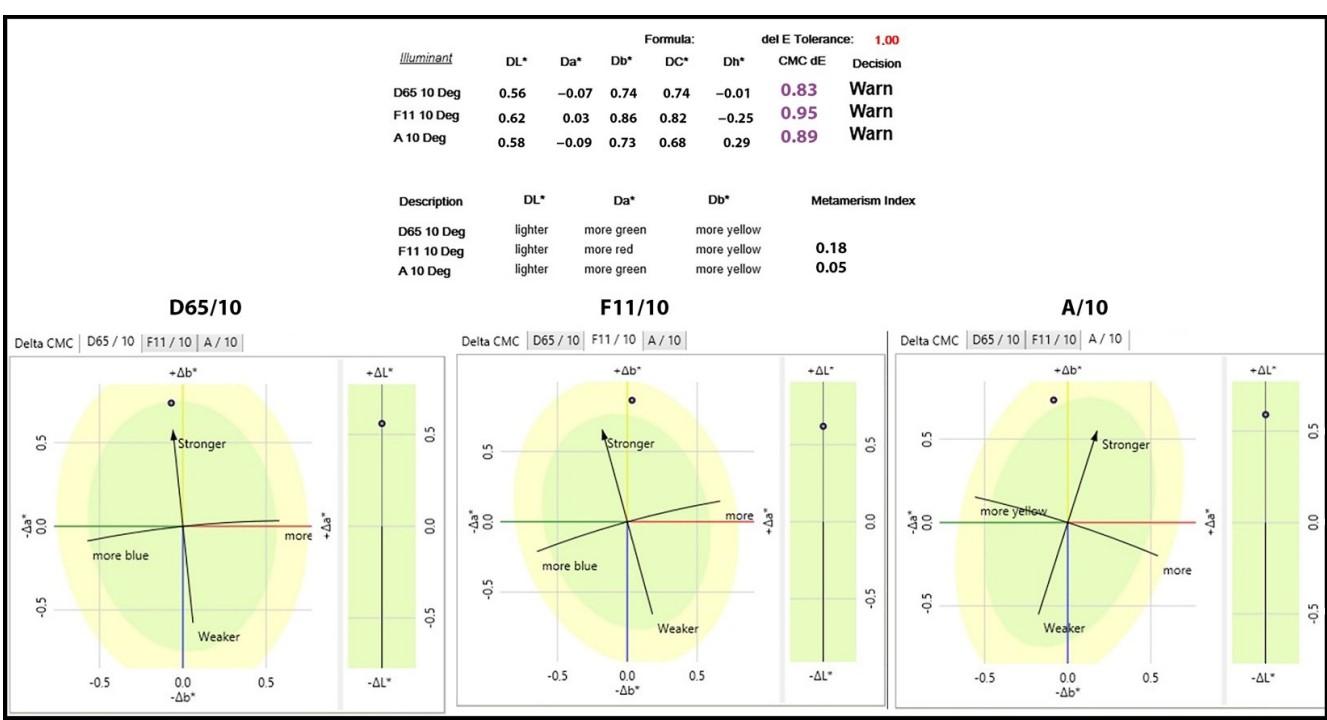

**Figure 16.** N-HEMP and B-HEMP in green colour.

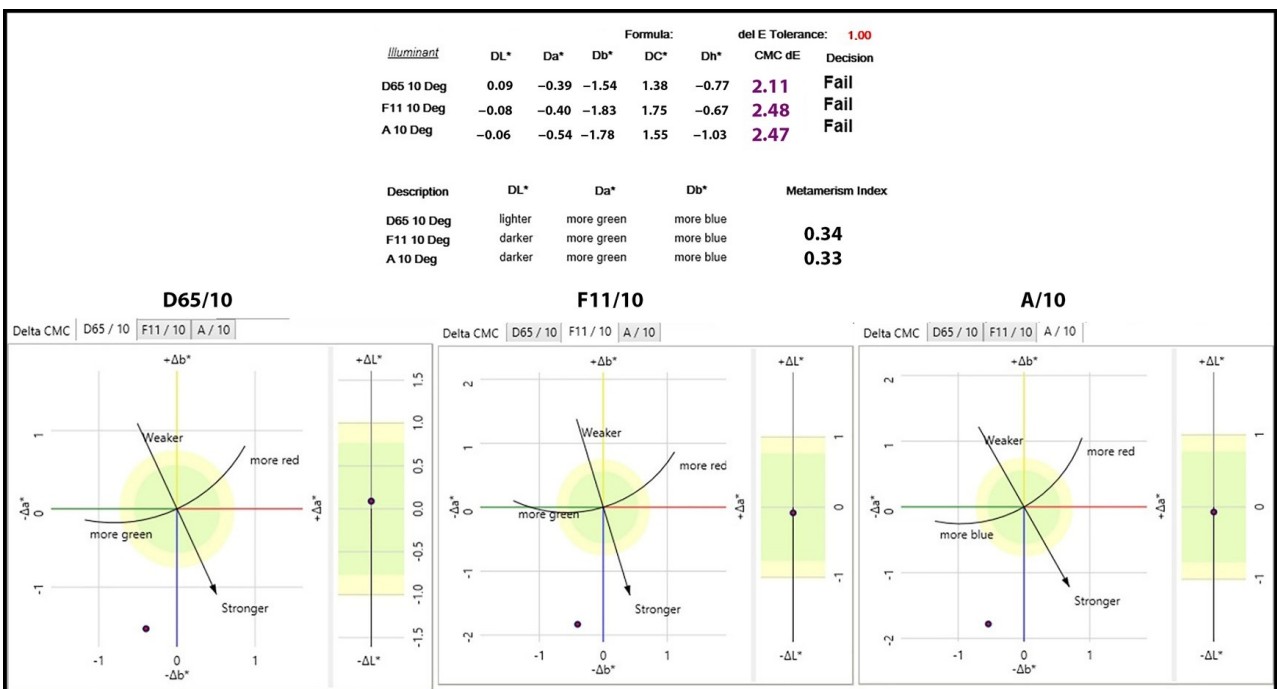

**Figure 17.** N-HEMP and B-HEMP in navy blue colour.

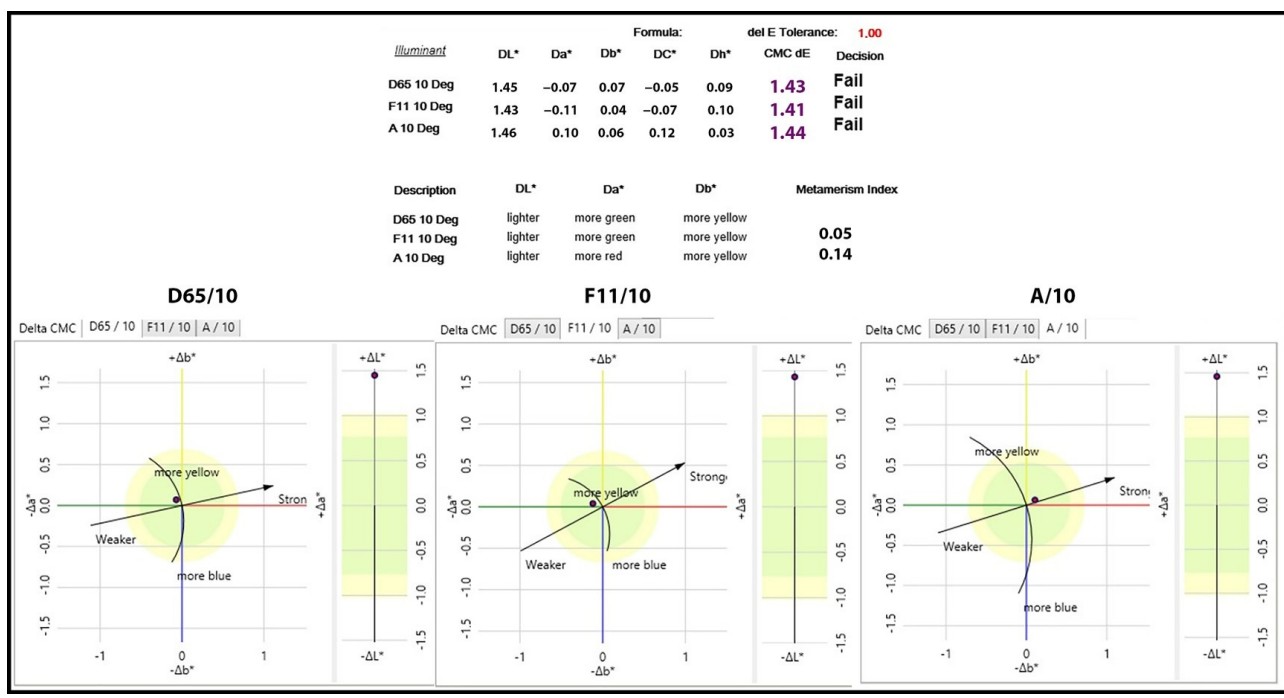

**Figure 18.** N-HEMP and B-HEMP in black colour.

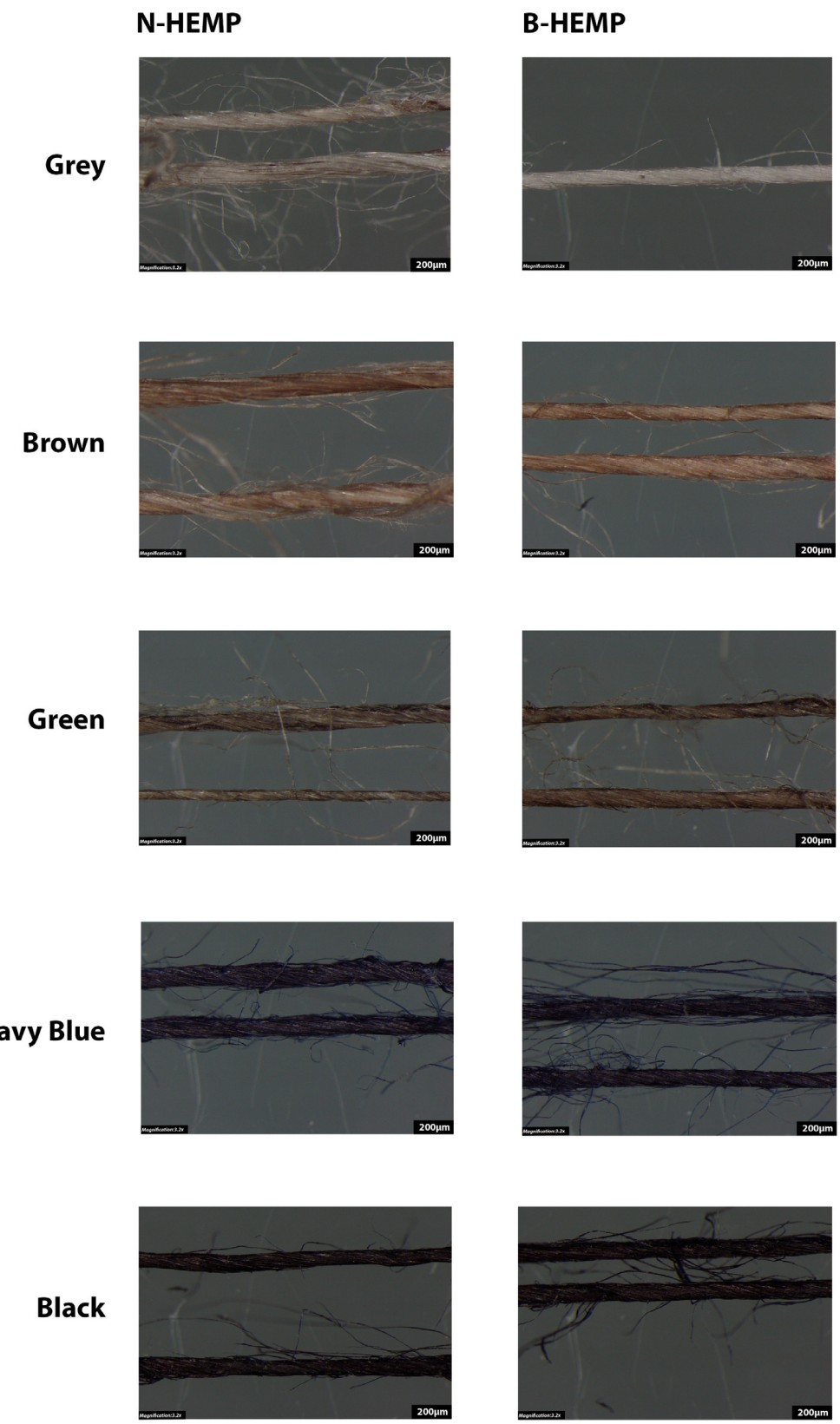

**Figure 19.** Visual Analysis of Dyed Yarns under Light Microscope.

## 4. Conclusions

The objective of this study was to collect information on the mechanical and physical properties, as well as the dyeability, of wet-spun hemp yarn in its natural and bleached

states to facilitate its use in the textile industry. The results indicate that bleach did not affect the yarn count, but had a positive effect on the yarn twist, as its value increased. On the other hand, bleach had a partially positive effect on the yarn's evenness, as it enhanced overall evenness by reducing thick and thin places, however, the process increased the neps and hairiness. In B-HEMP, the surface appearance test revealed lower values of thick and thin places, indicating a more homogenous and uniform structure. Under a microscope, images of yarn samples revealed the presence of nodes and crystalline regions on the fibres, with nodes having a larger diameter than crystalline regions. On the bleached yarn, microscopic images also reveal surface irregularities and chemical accumulation. The results of the yarn cross-section fibre count determination test indicate that the B-HEMP has a higher fibre density, signifying enhanced yarn performance. During the FTIR test, no differences in wavelength range and intensity were detected, suggesting that the two materials have the same chemical composition. The initial weight loss temperature of B-HEMP was marginally higher than that of N-HEMP, as determined by thermogravimetric analysis. Furthermore, the residual weight is less than that of N-HEMP. This demonstrates that bleaching improved the thermal stability of the yarn. The results of strength tests indicate that the tensile strength, elongation, and tenacity of hemp yarns increased after the bleaching process. This finding suggests that the bleaching process improves yarn's mechanical properties, resulting in a more durable structure. Throughout the strength tests performed on both unbleached and bleached wet-spun hemp yarns, various types of breakage points were identified. There were two distinct areas of weakness in the yarn: one where complete breakage occurred instantly and the other where gradual weakening led to strength loss. Due to the irregularities in the yarn, the strength test was repeated, with the yarn twisted into a double-ply. These findings suggest that double-plied yarns have a more uniform surface appearance, with less variation of thin and thick places. In comparison to the natural double-plied wet-spun hemp yarn, the bleached double-plied wet-spun hemp yarn showed superior maximum load and elongation at break values. However, the tenacity values were comparable. The bleached wet-spun hemp yarn has an increased overall strength. The evaluation of the strength of the dyed yarn revealed that dyed B-HEMP are superior to N-HEMP, particularly in the case of grey-coloured yarn. In contrast to natural wet-spun hemp yarns, alternative-coloured yarns have lower quality. Under varied lighting conditions, the grey colour values of bleached wet-spun hemp yarn and natural wet-spun hemp yarn differ according to colorimetric data analysis. The test revealed notable variations in colour results when dyeing wet-spun natural and bleached hemp yarns in brown under different lighting conditions, resulting in considerable variations in tone. The findings indicate that it is possible to produce hemp yarns in a green hue with uniform colouration, as there is no visual difference in terms of colour perception. Distinctive variations in colour between navy blue N-HEMP and B-HEMP can be observed by the analysis of their respective colour values, indicating a visually noticeable colour difference. Noticeable differences can be observed in the black-dyed hemp yarns. In summary, it can be concluded that the process of bleaching has had a positive effect on the wet-spun hemp yarn.

## 5. Future Work

Due to the limited data available on wet-spun hemp yarn, there are some significant research gaps. For instance, to enhance the properties of the wet-spun hemp yarn, the wet-spinning parameters can be studied for optimisation. Additionally, wet-spun hemp can be blended with other natural fibres making it much more suitable for use in apparel and home textiles. Similarly, eco-friendly dyed wet-spun hemp yarn can be tested for colour fastness and other properties. The cost of production of hemp fibres can be compared to existing fibres to ensure the commercial viability for mass production. This can be combined with a market and consumer study. Furthermore, significant research can be conducted on the life cycle assessment of the wet-spun hemp yarn to evaluate its environmental impact.

**Author Contributions:** S.T.: Conceptualization, formal analysis, resources, visualization. N.K.: Data curation, writing—original draft preparation. A.K.: Data curation, writing—original draft preparation. M.A.N.: Conceptualization, validation, writing—review and editing. A.N.: Writing—review and editing, visualization. F.T.: Writing—original draft preparation. M.U.: Conceptualization, methodology, validation, resources, writing—review and editing, project administration. All authors have read and agreed to the published version of the manuscript.

**Funding:** This research was funded by the University of Oradea within the framework of the grants competition "Scientific Research of Excellence Related to Priority Areas with Capitalization through Technology Transfer: INO-TRANSFER-UO-2nd Edition", Projects no. 236/2022.

**Institutional Review Board Statement:** Not applicable.

**Informed Consent Statement:** Not applicable.

**Data Availability Statement:** The data presented in this study are available on request from the corresponding author.

**Conflicts of Interest:** The authors declare no conflict of interest.

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
