# Peer review of "Analysing the Impact of the Bleaching Process on Wet Spun Hemp Yarn Properties"

_sustainability, doi:10.3390/su152416894_

Round 1

Reviewer 1 Report

Comments and Suggestions for Authors

The presented article is a study of the various characteristics of natural and bleached yarn from the hemp plant. The topic of the article is relevant and may be of interest to specialists and researchers in the fields of agriculture, plant growing and sustainable development. The article presents interesting experimental results obtained using various devices and instruments, but several points should be noted as recommendations and comments:

1. Authors should avoid multiple citations and consider each source separately. Authors should also include in the literature review modern and relevant research on the topic under consideration from international high-ranking journals.

2. What do the numbers 16-20 in Table 3 mean - the authors should clarify this. There are no links to figures in the text of the article - authors should add links to all figures. Figures 5, 6, 8, 12-16 should be enlarged. Table 10 may have mixed up the values in the right column. Figure 11 (caption) is repeated twice.

3. The description of the results in the article should be structured and titled by the authors, divided into subsections, etc.

4. What and how can one justify the improved strength characteristics of gray-dyed yarn?

5. Where and how do the authors plan to implement the results obtained in the work under review - this should be noted in the article.

6. Also, the authors should add a section “Directions for further research,” where the authors should describe planned work on the topic under consideration.

7. At the end of the article, authors should indicate the contribution of each author to the work done.

In general, the presented article leaves a positive impression, however, it is not without minor shortcomings. After eliminating these comments and taking into account the recommendations made, the presented article may be of interest to readers of the journal "Sustainability".

Comments on the Quality of English Language

Authors should carefully proofread the text of the article to avoid spelling and punctuation errors and also seek help from a native English speaker for final proofreading of the text.

Author Response

Attached as a PDF.

Reviewer 2 Report

Comments and Suggestions for Authors

1-this manuscript contain a good analysis but the results need clear discussion.

2-where is the chemical structure of hemp fibers in two form (bleached and natural).

3-at page 4 table 3 contain numbers 16-20 what is meaning of them ? also at table 3 the % depend on what weight of fabric or what?

4-at table 5 write the full name of Um and CVm under table.

5-at figure 1&2 you put 3 pictures A,B&C and you are not mention any think about them in discussion why?

6-where the discussion on figure 3,4,5 &6 .

7- when you discussed any figure you must write in text as shown at  figure * and write discussion where all figures from 1-17 not mentioned in text you mentioned only tables why?please write a comments for every figure 

8-at discussion of every properties put a tittle for every one for example "surface and morphology structure" then write as shown in figure 1 ....

9-you mentioned 10 samples and the results indicate to natural and bleached hump. why?

10-at figure 17 you put pictures for bleached and natural flax and this type of fiber not mentioned at all in materials or in all manuscript also at this figure you write a dyes(grey,green etc.) not mentioned in table 2 why?

11-Conclusions need to be more clear.

12- References need to be up to date and they are 25 not 26 .

13-check the grammar of this manuscript for example at page 9 line No 265 "table 9 indicates not indicate "

Comments on the Quality of English Language

check the language and grammar for all manuscript.

Author Response

Attached as a PDF.

Reviewer 3 Report

Comments and Suggestions for Authors

I give my opinion in the review document I am attaching.

​

Author Response

Attached as a PDF.

Round 2

Reviewer 2 Report

Comments and Suggestions for Authors

1-at figure 1 remove number 16 , the reference number put in text.

2-at figures 2&3 indicate to the thickness of N-HEMP and B-HEMP under microscope please mentioned the value of magnification for every picture.

3-at figure 11 which indicates to thickness of double plied samples under microscope, please clear under figure and in text the difference between picture A and B.

4-at figure 17 remove the sentence contain figure 39 its wrong . 
